# Consumption of High-Energy Food and Sugar Shows a Strong Positive Association with Low Mood in Control Subjects and Depressed Patients

**DOI:** 10.3390/nu17162594

**Published:** 2025-08-09

**Authors:** Tabita Dobai, Daniel Baksa, Xenia Gonda, Gabriella Juhasz, Nora Eszlari, Gyorgy Bagdy

**Affiliations:** 1Department of Pharmacodynamics, Faculty of Pharmaceutical Sciences, Semmelweis University, Nagyvárad tér 4, H-1089 Budapest, Hungary; dobai.tabita@stud.semmelweis.hu (T.D.); baksa.daniel@semmelweis.hu (D.B.); gonda.xenia@semmelweis.hu (X.G.); juhasz.gabriella@semmelweis.hu (G.J.); eszlari.nora@semmelweis.hu (N.E.); 2Center of Pharmacology and Drug Research & Development, Semmelweis University, Üllői Street 26, H-1085 Budapest, Hungary; 3Department of Personality and Clinical Psychology, Institute of Psychology, Faculty of Humanities and Social Sciences, Pazmany Peter Catholic University, Mikszáth Kálmán tér 1, H-1088 Budapest, Hungary; 4NAP3.0-SE Neuropsychopharmacology Research Group, Hungarian Brain Research Program, Semmelweis University, Üllői Street 26, H-1085 Budapest, Hungary; 5Department of Psychiatry and Psychotherapy, Semmelweis University, Gyulai Pál utca 2, H-1085 Budapest, Hungary; 6Department of Clinical Psychology, Semmelweis University, Üllői út 25, H-1091 Budapest, Hungary

**Keywords:** depression, sugar, carbohydrate, energy, anhedonia, BMI, food intake, general non-depressed population

## Abstract

**Background/Objectives**: Eating has been suggested to be one of the most important hedonic behaviors. Anhedonia, a symptom of depression, may be associated with decreased food intake, although increase of food intake could be a symptom of depression as well. Our aim was to explore the association of mood-related symptoms and anhedonia with carbohydrate and sugar intake in never-depressed control persons and depressed patients. **Methods**: In a large UK Biobank sample (>100,000), two-way regression models were constructed: first, for two lifetime depression variables (ICD-10 and CIDI), two current depression scores (PHQ-9 and a four-item score), and two anhedonia items as outcomes with 14 nutrient predictors, and then in the opposite direction, with nutrients as outcomes. **Results**: Energy density, free sugar, lactose, other sugars, and sucrose intake were higher, while fructose and glucose intake were lower in depressed patients compared to control subjects. Strong positive associations were found among energy measures, carbohydrate, free sugar, lactose, maltose, other sugars, and sucrose intake and almost all depression measures, including anhedonia. These associations were similar in the total sample and in the never-depressed control subjects as well. In contrast, fructose and glucose intake showed negative associations with the majority of the above measures. Sex, age, BMI, and Townsend deprivation index as predictors failed to show major effects on these associations. **Conclusions**: Our results suggest that consumption of high-energy food and sugar may be generally employed to alleviate mood disturbances and anhedonia in high-income countries by depressed patients and by never-depressed subjects, although the effects of sugars on depression cannot be ruled out.

## 1. Introduction

The average lifetime prevalence of depression is 12% [1], and it is predicted that it will represent the largest disease burden by 2030 [2]. While several depressogenic factors, including female sex and heritability, are non-modifiable, others, such as dietary habits, can be modified. The risk of depression is twice as high in obese persons compared to individuals of normal weight [3].

Generally, eating has been suggested to be one of the most important hedonic behaviors, and, presumably, people with depression are susceptible to consuming carbohydrates with the aim of self-soothing or self-ingratiating, as the rewarding role of eating sweet foods might result in temporary alleviation of low mood [4]. On the other hand, anhedonia (i.e., lack of motivation, interest, and pleasure), one of the two core symptoms of depression [5], correlates with a blunted reward sensitivity and shows an inconsistent association with food intake [6]. Loss of appetite can be interpreted as a critical anhedonic dimension of depression, and in animal models of depression decreased sucrose consumption, namely loss of preference for sucrose solution over drinking water, is a typical measurement of anhedonia-like behaviors [7]. At the same time, recent studies detected that people with self-reported anhedonia consumed more sweetened beverages compared to participants without anhedonia [8] and showed a more central intake of foods with a high or medium glycemic index [6], and that anhedonia predicted weight gain in the long term (over one year) [9].

According to recent meta-analyses, total sugar consumption increases the risk of depression, most likely in a dose-response manner [10,11,12]. However, these studies usually focused only on total or added sugar intake, mostly in beverages. According to our knowledge, only two studies investigated multiple sugar subtypes in association with depression, yielding conflicting results. A prospective study in the US [4] detected a positive association for added sugars and a negative association for lactose with self-reported depressive symptoms (measured by the short version of CES-D) in a sample of postmenopausal women. Glucose and sucrose intake showed elevated but non-significant odds ratios for depression in fully adjusted models. Furthermore, no relations were found between depression and total sugar, fructose, and starch intake. Additionally, higher consumption of nonjuice fruits and vegetables decreased the odds of depression. Another study in Japan [13] revealed a positive association for total sugars, free sugars, sucrose, glucose, and total fructose (but not lactose) with depressive symptoms (measured by CES-D) in young women and a negative association only for sucrose with depression in middle-aged females. These two studies suggest that different sugar types may associate differently with depression, and that these relationships might be influenced by cultural (e.g., dietary differences between Western and Eastern populations) and age-related factors. However, these investigations were restricted to females and used only a self-reported questionnaire to capture depression, namely the CES-D. Furthermore, no studies have addressed the relationship specifically between anhedonia and sugar subtypes.

On the other hand, the association between depression and carbohydrate intake may be more complex and bilateral in nature, leading to a vicious cycle [14], as depression is associated with increased craving for carbohydrates, as carbohydrates appear to improve mood and reduce depressive symptoms [15,16]. The effect direction of the depression–carbohydrate association is understudied, and, according to our knowledge, no study has addressed this topic by directly testing both effect directions.

Recently, the UK Biobank (UKB) provided novel, detailed dietary data, including consumption of sugar subtypes, giving us the opportunity to gain novel insight into the associations between depression/anhedonia and nutrient intake, including previously neglected general variables (such as energy density) and specific sugar subtypes (such as maltose). Thus, based on the above, our aim was to investigate bidirectional associations between depression and anhedonia and energy, carbohydrate, total sugar, and sugar subtype intake in a large sample of depressed patients and non-depressed controls.

## 2. Materials and Methods

### 2.1. Participants

The present study used UK Biobank data under application number 1602. Participant recruitment within the UK Biobank cohort has been detailed elsewhere [17]. Participants provided written informed consent, and all procedures were in accordance with the Declaration of Helsinki. Ethical approval for the study was given by the National Research Ethics Service Committee North West-Haydock (REC reference number: 21/NW/0157, dated 29 June 2021) [18]. For the present analyses, we included participants with no missing data for all of the phenotypes included in each regression model. For each of the nutrient variables, we included participants with no missing data for at least one of the five data acquisition instances (see Appendix A) of the Oxford WebQ diet questionnaire.

The sample sizes for analysis of each of the depression and anhedonia phenotypes were as follows: n = 210,058 for lifetime ICD-10 depression; n = 110,627 for lifetime CIDI depression; n = 209,615 for the four-item current depression score; n = 113,234 for the PHQ9 current depression score; n = 204,935 for the anhedonia item of the four-item depression questionnaire; and n = 112,984 for the anhedonia item of the PHQ9 depression questionnaire.

Analyses within separate groups of never-depressed and ever-depressed participants according to ICD-10 depression phenotype (the binary depression variable with the higher sample size) were carried out, with sample sizes as follows: n = 189,873 never-depressed and n = 19,742 ever-depressed participants for the four-item current depression score; n = 102,965 never-depressed and n = 10,269 ever-depressed participants for the PHQ9 current depression score; n = 185,821 never-depressed and n = 19,114 ever-depressed participants for the anhedonia item of the four-item depression questionnaire; and n = 102,750 never-depressed and n = 10,234 ever-depressed participants for the anhedonia item of the PHQ9 depression questionnaire.

### 2.2. Study Variables

Our study variables, with their UK Biobank data field identifiers, data acquisition methods, and instances, in order by data acquisition date, are detailed in Appendix A.

#### 2.2.1. Depression Phenotypes

In order to widen former research perspectives to subclinical levels of depression, as well as to future possibilities of intervention, primary prevention, or relapse prevention for diagnosed major depression, our aim was to cover depression with distinct measures available in the UK Biobank. In particular, we used two different phenotypes reflecting a lifetime diagnosis of depression, two phenotypes reflecting current depression symptom sum scores, and two anhedonia measures from each of the two sets of current depression symptom items.

ICD-10 (International Classification of Diseases, Tenth Revision) lifetime diagnosis of either a depressive episode (F32, data-field ‘130894’) or recurrent depressive disorder (F33, data-field ‘130896’) was used as a binary measure [19].

CIDI (Composite International Diagnostic Interview) lifetime depression, used similarly as a binary measure, was assessed by the method described by Cai et al., 2020 [20], as detailed in Appendix A in their publication. However, to maximize the sample size, we applied a refined approach to determine a CIDI depression diagnosis for participants with missing questionnaire items. Rather than excluding individuals with one or more missing responses, we assessed whether the available data were sufficient to ascertain the diagnosis with certainty. If a participant’s responses provided enough information to conclusively establish the presence or absence of depression according to CIDI criteria, their data were retained in the analysis. This strategy ensured the inclusion of a larger subset of participants while maintaining diagnostic accuracy.

In addition to these two tools assessing a lifetime depression diagnosis, we used two continuous measures assessing the severity of current depressive symptoms, in order to cover the entire spectrum of depression, involving also subclinical levels that do not meet the clinical requirements of a depressive episode. These two measures of current depressive symptom severity were not used for diagnosis.

The four-item current depression score [21,22] assessed the presence of depressed mood, anhedonia (loss of interest), restlessness, and lack of energy over the past 2 weeks and was calculated by summing individual scores (1–4) for the four items detailed in Appendix A, divided by the number of items responded to by the respective participant, and multiplied by 4, thus yielding a score ranging from 4 to 16.

The Patient Health Questionnaire 9 (PHQ9) also assessed the presence of depressive symptoms (also including anhedonia) over the last 2 weeks. The PHQ9 score [23,24] was calculated by summing individual scores (1–4) for the nine items detailed in Appendix A, divided by the number of items responded to by the respective participant, and multiplied by 9, thus yielding a score ranging from 9 to 36.

The anhedonia item of the four-item current depression measurement was scored from 1 (“not at all”) to 4 (“nearly every day”) for the following question: “Over the past two weeks, how often have you had little interest or pleasure in doing things?”.

The anhedonia item of the PHQ9 questionnaire was scored as 1 (“not at all”) to 4 (“nearly every day”) for the following question: “Over the last 2 weeks, how often have you been bothered by any of the following problems? [depressive symptoms] Little interest or pleasure in doing things”.

#### 2.2.2. Nutrient Variables

Dietary intake data were collected using version 1.4 of the UK Biobank Oxford WebQ questionnaire [25], released on 7 March 2024, which recorded participants’ eating habits via a 24 h recall survey. Nutrient intake was estimated using the UK Nutrient Databank (UKNDB, 2013), which was developed as part of the National Diet and Nutrition Survey (NDNS) [26] and provided the food composition data for nutrient calculations [27]. The UKNDB was used because its years 5–6 data collection period closely aligned with the dietary data collection period for the Oxford WebQ (2009–2012). The UKNDB database includes nutritional information for over 5600 food items, of which 681 food codes were incorporated into the Oxford WebQ questionnaire.

Nutrient variables were derived from these dietary data and categorized in the UK Biobank under “Estimated food nutrients yesterday”, which comprises 63 variables in total, of which 14 were selected for our analyses. Definitions for the nutrient variables were based on the descriptions provided in the UK Biobank Data Showcase, while the classification of carbohydrate types was based on the definitions from the Scientific Advisory Committee on Nutrition (SACN) (https://assets.publishing.service.gov.uk/government/uploads/system/uploads/attachment_data/file/445503/SACN_Carbohydrates_and_Health.pdf, accessed on 5 April 2025), as reported by Perez-Cornago et al. [27].

Each analysis used the nutrient variable value averaged among all data acquisition instances answered by the respective participant, out of the five instances listed in Appendix A.

The following nutrient indicators were included in our analyses, as defined in the UK Biobank Data Showcase and NDNS documentation:Energy (kJ): Total dietary energy consumed in the previous 24 h, including energy from alcohol (UK Biobank Data Showcase).Energy density (kJ/g): Total energy divided by the weight of food consumed, excluding beverages [27].Energy from beverages (kJ): Energy derived exclusively from beverages (UK Biobank Data Showcase).Total weight of beverages only (g): Sum of all liquid intake over 24 h (UK Biobank Data Showcase).Carbohydrates (g): “Major source of energy in the diet and include a range of compounds all containing carbon, hydrogen and oxygen” (https://assets.publishing.service.gov.uk/government/uploads/system/uploads/attachment_data/file/445503/SACN_Carbohydrates_and_Health.pdf, p. 10, accessed on 5 April 2025). Based on varying linkages and chain lengths, this group includes a wide spectrum of compounds, from mono- and disaccharides (commonly referred to as sugars) to oligo- and polysaccharides such as dietary fiber (SACN).Free sugars (g): “Include monosaccharides and disaccharides added to foods and beverages by the manufacturer, cook or consumer, and sugars naturally present in honey, syrups, fruit juices and fruit juice concentrates” ((https://www.who.int/news/item/04-03-2015-who-calls-on-countries-to-reduce-sugars-intake-among-adults-and-children, accessed on 5 April 2025); https://assets.publishing.service.gov.uk/government/uploads/system/uploads/attachment_data/file/445503/SACN_Carbohydrates_and_Health.pdf, p. 18, accessed on 5 April 2025).Other sugars (g): Composed mainly of oligosaccharides, where data are available (NDNS Years 5–6 2012/13–2013/14 User Guide for UK Data).Intrinsic and milk sugars (g): Naturally present sugars within intact cellular structures or in milk (SACN).Non-milk extrinsic sugars (NMESs, g): “Sugars not contained within the cellular structure of a food, except lactose in milk and milk products” (https://assets.publishing.service.gov.uk/government/uploads/system/uploads/attachment_data/file/445503/SACN_Carbohydrates_and_Health.pdf, p. 18, accessed on 5 April 2025). This category includes 50% of sugars from dried, stewed, or canned fruits, whereas free sugars exclude these.Fructose (g), glucose (g), lactose (g), maltose (g), sucrose (g): Individual sugars defined by their chemical composition and natural sources.

ChatGPT-4o was used for to draft and polish this section.

#### 2.2.3. Potential Confounding Variables

In our regression analyses, we included sex, age, body mass index (BMI), and Townsend index of socioeconomic deprivation as covariates. The relationships of depression to sex [28], BMI [29], and Townsend index [30] have been well established. Moreover, one study reported that age, BMI, and indices of socioeconomic status are significantly associated with both glycemic index and incident depression 3 years later [4].

### 2.3. Analyses

Data cleaning and preparation were performed in R 4.1.2.

Statsmodels 0.14.4 and Patsy 0.5.6 were used for the regression analyses, and Pandas 2.2.3, Scipy 1.14.1, and Numpy 2.1.2 were used for the descriptive statistics calculations, all within Python 3.11.9. ChatGPT-4o was used for support in Python script writing.

Without accounting for the shared variance with any of the potential confounding variables, the Mann–Whitney U test was used to compare the distribution of each of the 14 nutrient intake variables between the ICD-10 ever-depressed subgroup and the never-depressed subgroup. Because of the 14 tests, a Bonferroni-corrected *p* ≤ 3.57 × 10^−3^ significance threshold was applied.

Logit regression models were constructed for the two lifetime depression diagnosis binary variables as outcomes, and ordinary least squares regression models were constructed for the two current depression scores, as well as for the two one-item anhedonia measures as outcomes. Each of the 14 nutrient variables was used as a predictor in separate models, while sex, age, BMI, and Townsend deprivation index were included as predictors in every model. Regression models for the two current depression scores and the two anhedonia items as outcomes were also constructed separately for the ICD-10 ever-depressed subgroup and the never-depressed subgroup. The 14 nutrient variables as predictors and the 14 outcome variables yielded a Bonferroni-corrected *p* ≤ 2.55 × 10^−4^ significance threshold.

For validation purposes, and because of the well-established association between depression and all of the potential confounding variables in our present study (see Section 2.2.3. for details), the same regression models were carried out with distinct combinations of predictor variables in addition to the respective nutrient variable (Appendix A), as follows:sex, age;sex, age, BMI;sex, age, Townsend index.

The same Bonferroni correction was applied within each of these three sets of regression models (Appendix A).

To test associations between depression and nutrient intake from the other direction, ordinary least squares regression models were constructed for the 14 nutrient variables as outcomes. Sex, age, BMI, and Townsend deprivation index were predictors in every model, in addition to each of the two lifetime depression diagnosis binary variables, the two current depression scores, and the two one-item anhedonia measures, each in a separate model. Regression models with each of the two current depression scores and the two one-item anhedonia measures were also carried out separately within the ICD-10 ever-depressed subgroup and the never-depressed subgroup. The 14 nutrient variables as outcomes and the 14 different combinations of depression predictors and whole sample/subgroup yielded a Bonferroni-corrected *p* ≤ 2.55 × 10^−4^ significance threshold.

## 3. Results

### 3.1. Summary of the Results

An overview of all results, including differences in nutrient intake between patients and controls, as well as bidirectional associations between depression and anhedonia and energy, carbohydrate, total sugar, and sugar subtype intake in the total sample, in never-depressed controls, and in ever-depressed patients is given in Table 1. Descriptive statistics are collected in Table 2 and Table 3, while details of the analyses supporting Table 1 (means, SDs, SEMs, Mann–Whitney U values, *p* values, coefficients) are given in Table 4, Table 5, Table 6, Table 7, Table 8, Table 9 and Table 10.

### 3.2. Descriptive Statistics

Descriptive statistics for categorical variables (Table 2) and for continuous variables (Table 3) were calculated for those participants having no missing data for all questions regarding sex, age, BMI, Townsend index, and nutrient variables derived from the Oxford WebQ questionnaire.

**Table 2 nutrients-17-02594-t002:** Descriptive statistics for the categorical variables in our study.

Variable Name		Count
Sex	Female	115,751
	Male	94,307
ICD-10 F32 or F33	Never-depressed controls	190,263
	Ever-depressed patients	19,795
CIDI lifetime depression	Never-depressed controls	94,308
	Ever-depressed patients	16,319
	Missing data	99,431

ICD: International Classification of Diseases, F32 or F33: lifetime diagnosis of either a depressive episode or recurrent depressive disorder, CIDI: Composite International Diagnostic Interview.

**Table 3 nutrients-17-02594-t003:** Descriptive statistics for the continuous variables in our study.

Categories	Name	Unit	Mean	SD	SEM
Potential confounding variables	Age	Years	56.083	7.9485	0.0173
	Townsend	Higher score means higher material deprivation	−1.572	2.8763	0.0063
	BMI	kg/m^2^	26.958	4.6577	0.0102
Depression phenotypes	Four-item current depression score	Likert scale	5.497	1.9793	0.0043
	PHQ9 score	Likert scale	11.757	3.6712	0.0109
	Anhedonia item of the four-item current depression questionnaire	Likert scale	1.237	0.5579	0.0012
	Anhedonia item of the PHQ9 questionnaire	Likert scale	1.238	0.5646	0.0017
Nutrient variables	Energy	kJ	8643.222	2539.7981	5.5415
	Energy density	kJ/g	6.477	1.6925	0.0037
	Energy from beverages	kJ	1324.795	936.3678	2.0430
	Total weight of beverages only	g	2036.626	664.4986	1.4499
	Carbohydrate	g	254.463	81.2593	0.1773
	Free sugar	g	60.897	36.8552	0.0804
	Fructose	g	28.183	14.8827	0.0325
	Glucose	g	26.415	13.2198	0.0288
	Intrinsic and milk sugars	g	60.138	27.7705	0.0606
	Lactose	g	13.936	7.7924	0.0170
	Maltose	g	6.762	7.0043	0.0153
	Non-milk extrinsic sugars	g	65.054	37.3275	0.0814
	Other sugars	g	2.348	3.0090	0.0066
	Sucrose	g	47.507	25.7075	0.0561

SD: standard deviation, SEM: standard error of mean, BMI: body mass index, PHQ9: Patient Health Questionnaire 9, kg: kilograms, m^2^: square meters, kJ: kilojoules, g: grams.

### 3.3. Nutrient Intake Differences Between ICD-10 Never-Depressed Controls and Ever-Depressed Patients

According to our results, all nutrient variables except for intake of carbohydrates, as well as intake of intrinsic and milk sugars, showed a significant difference between the ICD-10 ever-depressed patients and never-depressed controls (Table 4, see also Table 1). In the case of energy, energy from beverages, fructose, glucose, and maltose, never-depressed controls showed a higher mean level of intake compared to ever-depressed patients. In contrast, the mean level of intake of energy density, total weight of beverages only, free sugar, lactose, non-milk extrinsic sugars, and other sugars, as well as sucrose, was higher in ever-depressed patients compared to never-depressed controls (Table 4, see also Table 1). Some of these between-subgroup differences were considerably high, with no overlap in the 95% confidence interval of mean of the two subgroups (based on the standard error of the mean multiplied by 1.96) (Table 4, see also Table 1). In particular, never-depressed controls showed a considerably higher mean intake of maltose and energy from beverages compared to ever-depressed patients, while ever-depressed patients showed a considerably higher mean intake of energy density, total weight of beverages only, free sugar, lactose, non-milk extrinsic sugars, other sugars, and sucrose compared to never-depressed controls.

**Table 4 nutrients-17-02594-t004:** Comparison of nutrient intake between never-depressed controls and ever-depressed patients.

Nutrient Variables	Never-Depressed Controls			Ever-Depressed Patients			Mann–Whitney U	*p* Value
	Mean	SD	SEM	Mean	SD	SEM		
Energy	8644.416	2516.6618	5.7583	8611.313	2758.1716	19.5502	1,864,265,736	7.278 × 10^−6^ **
Energy density	6.466	1.6750	0.0038	6.578	1.8555	0.0132	1,948,933,477	4.302 × 10^−9^ **
Energy from beverages	1326.168	932.4856	2.1336	1307.553	972.5777	6.8937	1,851,706,843	1.726 × 10^−9^ **
Total weight of beverages only	2032.054	660.2766	1.5108	2077.920	702.1125	4.9766	1,966,165,503	1.461 × 10^−15^ **
Carbohydrate	254.286	80.4939	0.1842	255.732	88.5589	0.6277	1,899,687,875	8.791 × 10^−1^
Free sugar	60.651	36.4080	0.0833	63.192	41.0374	0.2909	1,935,099,078	2.918 × 10^−5^ **
Fructose	28.203	14.7787	0.0338	27.943	15.8569	0.1124	1,855,986,409	3.837 × 10^−8^ **
Glucose	26.421	13.1155	0.0300	26.309	14.1874	0.1006	1,866,602,588	2.675 × 10^−5^ **
Intrinsic and milk sugars	60.121	27.5549	0.0630	60.188	29.8092	0.2113	1,880,890,109	1.422 × 10^−2^
Lactose	13.886	7.7367	0.0177	14.365	8.3061	0.0589	1,953,236,826	1.568 × 10^−10^ **
Maltose	6.785	7.0054	0.0160	6.503	6.9382	0.0492	1,844,385,122	4.600 × 10^−12^ **
Non-milk extrinsic sugars	64.810	36.8693	0.0844	67.299	41.5893	0.2948	1,933,828,805	5.713 × 10^−5^ **
Other sugars	2.323	2.9498	0.0067	2.589	3.5212	0.0250	1,959,068,790	1.143 × 10^−12^ **
Sucrose	47.271	25.3765	0.0581	49.744	28.7381	0.2037	1,973,103,569	1.057 × 10^−18^ **

The mean, SD, and SEM of each nutrient intake variable are displayed within the separate subgroups of ICD-10 F32 or F33 never-depressed controls and ever-depressed patients, as well as the Mann–Whitney U test statistic and its *p*-value for the difference of the nutrient variable’s distribution between these two subgroups. SD: standard deviation, SEM: standard error of mean, ICD: International Classification of Diseases, F32 or F33: lifetime diagnosis of either a depressive episode or recurrent depressive disorder, **: significant at the Bonferroni-corrected *p* ≤ 3.57 × 10^−3^ significance threshold.

### 3.4. Relevance of Nutrient Variables in Depression and Anhedonia Outcomes

The *p*-value and unstandardized regression coefficient of each nutrient variable as a predictor in separate regression models are detailed for the two lifetime depression diagnoses (Table 5, see also Table 1), for the two current depression scores (Table 6, see also Table 1), and for the two variables of current anhedonia level (Table 7, see also Table 1) as outcomes.

The results of additional regression models for the same outcome variables but with distinct combinations of predictor variables are shown in Appendix A, showing that our present results were not remarkably modified by these distinct combinations of the potential confounding variables (sex, age, BMI, and Townsend deprivation index) that have a well-established link to depression.

**Table 5 nutrients-17-02594-t005:** Results of the regression models of nutrient intake for lifetime depression as the outcome.

Predictor in the Regression Model	ICD-10 F32 or F33		CIDI Lifetime Depression	
	*p* Value	Coefficient	*p* Value	Coefficient
Energy	5.133 × 10^−9^ **	1.747 × 10^−5^	2.270 × 10^−3^	1.168 × 10^−5^
Energy density	5.226 × 10^−24^ **	4.484 × 10^−2^	4.171 × 10^−1^	4.566 × 10^−3^
Energy from beverages	2.230 × 10^−8^ **	4.610 × 10^−5^	2.010 × 10^−1^	1.330 × 10^−5^
Total weight of beverages only	3.158 × 10^−27^ **	1.203 × 10^−4^	2.102 × 10^−16^ **	1.140 × 10^−4^
Carbohydrate	1.265 × 10^−19^ **	8.256 × 10^−4^	1.026 × 10^−6^ **	5.744 × 10^−4^
Free sugar	2.751 × 10^−48^ **	2.838 × 10^−3^	1.289 × 10^−8^ **	1.454 × 10^−3^
Fructose	2.411 × 10^−1^	5.890 × 10^−4^	1.209 × 10^−2^	1.559 × 10^−3^
Glucose	1.086 × 10^−3^	1.831 × 10^−3^	2.305 × 10^−4^ **	2.574 × 10^−3^
Intrinsic and milk sugars	5.545 × 10^−3^	7.437 × 10^−4^	3.196 × 10^−8^ **	1.801 × 10^−3^
Lactose	4.186 × 10^−29^ **	1.042 × 10^−2^	1.000 × 10^−12^ **	8.124 × 10^−3^
Maltose	3.702 × 10^−4^	4.171 × 10^−3^	3.553 × 10^−1^	1.380 × 10^−3^
Non-milk extrinsic sugars	1.046 × 10^−47^ **	2.777 × 10^−3^	8.230 × 10^−10^ **	1.541 × 10^−3^
Other sugars	1.798 × 10^−18^ **	1.901 × 10^−2^	2.145 × 10^−8^ **	1.574 × 10^−2^
Sucrose	2.090 × 10^−59^ **	4.440 × 10^−3^	2.608 × 10^−17^ **	3.028 × 10^−3^

Logit regression models were constructed for the two lifetime depression diagnosis binary variables as outcomes, with each nutrient variable as a predictor in separate models, in addition to sex, age, body mass index, and Townsend index of socioeconomic deprivation as additional predictors in every model. **: significant at the Bonferroni-corrected *p* ≤ 2.55 × 10^−4^ significance threshold, ICD: International Classification of Diseases, F32 or F33: lifetime diagnosis of either a depressive episode or recurrent depressive disorder, CIDI: Composite International Diagnostic Interview.

**Table 6 nutrients-17-02594-t006:** Results of the regression models of nutrient intake for current depression as the outcome.

Predictor in the Regression Model	Four-Item Current Depression Score						PHQ9 Score					
	Total Sample		Never-Depressed Controls		Ever-Depressed Patients		Total Sample		Never-Depressed Controls		Ever-Depressed Patients	
	*p* Value	Coefficient	*p* Value	Coefficient	*p* Value	Coefficient	*p* Value	Coefficient	*p* Value	Coefficient	*p* Value	Coefficient
Energy	7.797 × 10^−64^ **	2.912 × 10^−5^	1.438 × 10^−62^ **	2.754 × 10^−5^	3.955 × 10^−2^	1.539 × 10^−5^	2.466 × 10^−26^ **	5.078 × 10^−5^	1.012 × 10^−22^ **	4.449 × 10^−5^	5.304 × 10^−3^	6.301 × 10^−5^
Energy density	1.266 × 10^−231^ **	8.447 × 10^−2^	3.873 × 10^−170^ **	6.915 × 10^−2^	1.186 × 10^−34^ **	1.372 × 10^−1^	1.283 × 10^−86^ **	1.403 × 10^−1^	2.636 × 10^−72^ **	1.215 × 10^−1^	2.170 × 10^−8^ **	1.857 × 10^−1^
Energy from beverages	5.085 × 10^−9^ **	2.739 × 10^−5^	8.860 × 10^−6^ **	1.980 × 10^−5^	6.274 × 10^−2^	3.938 × 10^−5^	1.007 × 10^−6^ **	6.244 × 10^−5^	2.609 × 10^−3^	3.635 × 10^−5^	2.452 × 10^−3^	1.869 × 10^−4^
Total weight of beverages only	6.804 × 10^−17^ **	5.383 × 10^−5^	5.266 × 10^−9^ **	3.583 × 10^−5^	1.486 × 10^−1^	4.158 × 10^−5^	5.304 × 10^−16^ **	1.416 × 10^−4^	2.219 × 10^−6^ **	7.831 × 10^−5^	1.769 × 10^−5^ **	3.597 × 10^−4^
Carbohydrate	2.255 × 10^−111^ **	1.194 × 10^−3^	4.186 × 10^−98^ **	1.070 × 10^−3^	1.089 × 10^−4^ **	8.912 × 10^−4^	2.054 × 10^−30^ **	1.688 × 10^−3^	1.307 × 10^−21^ **	1.334 × 10^−3^	1.262 × 10^−3^	2.242 × 10^−3^
Free sugar	6.335 × 10^−218^ **	3.701 × 10^−3^	4.588 × 10^−157^ **	3.008 × 10^−3^	5.439 × 10^−22^ **	4.797 × 10^−3^	4.585 × 10^−91^ **	6.543 × 10^−3^	6.367 × 10^−60^ **	5.025 × 10^−3^	2.015 × 10^−13^ **	1.094 × 10^−2^
Fructose	1.806 × 10^−10^ **	−1.817 × 10^−3^	3.372 × 10^−9^ **	−1.605 × 10^−3^	3.141 × 10^−4^	−4.554 × 10^−3^	1.006 × 10^−5^ **	−3.416 × 10^−3^	1.059 × 10^−7^ **	−3.890 × 10^−3^	8.371 × 10^−1^	−7.662 × 10^−4^
Glucose	5.928 × 10^−2^	−6.049 × 10^−4^	3.972 × 10^−2^	−6.288 × 10^−4^	2.524 × 10^−2^	−3.159 × 10^−3^	3.729 × 10^−1^	−7.773 × 10^−4^	3.603 × 10^−2^	−1.731 × 10^−3^	5.757 × 10^−1^	2.348 × 10^−3^
Intrinsic and milk sugars	3.670 × 10^−8^ **	−8.445 × 10^−4^	1.495 × 10^−7^ **	−7.681 × 10^−4^	1.922 × 10^−4^ **	−2.520 × 10^−3^	9.183 × 10^−4^	−1.361 × 10^−3^	4.363 × 10^−6^ **	−1.786 × 10^−3^	9.502 × 10^−1^	1.223 × 10^−4^
Lactose	4.248 × 10^−25^ **	5.625 × 10^−3^	1.629 × 10^−13^ **	3.822 × 10^−3^	3.647 × 10^−2^	5.033 × 10^−3^	7.955 × 10^−8^ **	7.683 × 10^−3^	1.046 × 10^−2^	3.478 × 10^−3^	3.478 × 10^−2^	1.419 × 10^−2^
Maltose	1.101 × 10^−13^ **	4.816 × 10^−3^	1.029 × 10^−12^ **	4.368 × 10^−3^	1.938 × 10^−1^	4.001 × 10^−3^	5.892 × 10^−14^ **	1.355 × 10^−2^	5.038 × 10^−11^ **	1.117 × 10^−2^	1.022 × 10^−2^	2.327 × 10^−2^
Non-milk extrinsic sugars	8.700 × 10^−202^ **	3.500 × 10^−3^	3.307 × 10^−144^ **	2.832 × 10^−3^	4.794 × 10^−20^ **	4.484 × 10^−3^	6.200 × 10^−86^ **	6.243 × 10^−3^	8.997 × 10^−56^ **	4.757 × 10^−3^	5.712 × 10^−13^ **	1.056 × 10^−2^
Other sugars	6.583 × 10^−42^ **	1.905 × 10^−2^	1.144 × 10^−29^ **	1.532 × 10^−2^	5.930 × 10^−4^	1.953 × 10^−2^	1.264 × 10^−39^ **	5.030 × 10^−2^	1.117 × 10^−28^ **	4.066 × 10^−2^	1.689 × 10^−5^ **	7.147 × 10^−2^
Sucrose	1.309 × 10^−278^ **	5.900 × 10^−3^	6.577 × 10^−200^ **	4.787 × 10^−3^	1.279 × 10^−28^ **	7.773 × 10^−3^	1.070 × 10^−111^ **	1.023 × 10^−2^	2.877 × 10^−72^ **	7.806 × 10^−3^	6.668 × 10^−17^ **	1.730 × 10^−2^

Ordinary least squares regression models were constructed for the two current depression scores as outcomes, with each nutrient variable as a predictor in separate models, in addition to sex, age, body mass index, and Townsend index of socioeconomic deprivation as additional predictors in every model. Never-depressed and ever-depressed subgroups are defined according to the ICD-10 F32 or F33. **: significant at the Bonferroni-corrected *p* ≤ 2.55 × 10^−4^ significance threshold, PHQ9: Patient Health Questionnaire 9, ICD: International Classification of Diseases, F32 or F33: lifetime diagnosis of either a depressive episode or recurrent depressive disorder.

**Table 7 nutrients-17-02594-t007:** Results of the regression models of nutrient intake for anhedonia as the outcome.

Predictor in the Regression Model	Anhedonia Item of the Four-Item Current Depression Questionnaire						Anhedonia Item of the PHQ9 Questionnaire					
	Total Sample		Never-Depressed Controls		Ever-Depressed Patients		Total Sample		Never-Depressed Controls		Ever-Depressed Patients	
	*p* Value	Coefficient	*p* Value	Coefficient	*p* Value	Coefficient	*p* Value	Coefficient	*p* Value	Coefficient	*p* Value	Coefficient
Energy	7.059 × 10^−13^ **	3.588 × 10^−6^	1.053 × 10^−10^ **	3.130 × 10^−6^	6.500 × 10^−1^	9.943 × 10^−7^	6.870 × 10^−9^ **	4.332 × 10^−6^	5.367 × 10^−7^ **	3.644 × 10^−6^	1.416 × 10^−1^	5.028 × 10^−6^
Energy density	2.014 × 10^−115^ **	1.718 × 10^−2^	1.753 × 10^−71^ **	1.305 × 10^−2^	1.827 × 10^−25^ **	3.419 × 10^−2^	1.217 × 10^−60^ **	1.827 × 10^−2^	6.105 × 10^−48^ **	1.576 × 10^−2^	2.226 × 10^−7^ **	2.599 × 10^−2^
Energy from beverages	5.945 × 10^−13^ **	9.760 × 10^−6^	3.812 × 10^−9^ **	7.701 × 10^−6^	1.084 × 10^−2^	1.579 × 10^−5^	4.479 × 10^−8^ **	1.092 × 10^−5^	5.066 × 10^−5^ **	7.844 × 10^−6^	6.095 × 10^−3^	2.559 × 10^−5^
Total weight of beverages only	2.635 × 10^−5^ **	7.832 × 10^−6^	5.733 × 10^−2^	3.422 × 10^−6^	3.771 × 10^−1^	7.431 × 10^−6^	1.419 × 10^−5^ **	1.186 × 10^−5^	2.938 × 10^−2^	5.778 × 10^−6^	5.225 × 10^−2^	2.461 × 10^−5^
Carbohydrate	1.691 × 10^−30^ **	1.770 × 10^−4^	1.641 × 10^−23^ **	1.495 × 10^−4^	1.676 × 10^−1^	9.308 × 10^−5^	2.957 × 10^−11^ **	1.531 × 10^−4^	1.595 × 10^−7^ **	1.174 × 10^−4^	1.286 × 10^−1^	1.599 × 10^−4^
Free sugar	6.473 × 10^−104^ **	7.367 × 10^−4^	1.287 × 10^−66^ **	5.709 × 10^−4^	1.824 × 10^−12^ **	1.027 × 10^−3^	4.778 × 10^−52^ **	7.675 × 10^−4^	4.337 × 10^−35^ **	6.099 × 10^−4^	8.940 × 10^−7^ **	1.107 × 10^−3^
Fructose	2.145 × 10^−12^ **	−5.786 × 10^−4^	5.360 × 10^−10^ **	−4.943 × 10^−4^	3.635 × 10^−5^ **	−1.526 × 10^−3^	2.123 × 10^−8^ **	−6.773 × 10^−4^	5.099 × 10^−9^ **	−6.855 × 10^−4^	1.477 × 10^−1^	−8.170 × 10^−4^
Glucose	3.781 × 10^−4^	−3.295 × 10^−4^	6.932 × 10^−4^	−3.042 × 10^−4^	3.757 × 10^−3^	−1.197 × 10^−3^	2.323 × 10^−3^	−4.153 × 10^−4^	2.406 × 10^−4^ **	−4.859 × 10^−4^	4.827 × 10^−1^	−4.461 × 10^−4^
Intrinsic and milk sugars	1.416 × 10^−9^ **	−2.680 × 10^−4^	5.762 × 10^−8^ **	−2.323 × 10^−4^	4.462 × 10^−5^ **	−8.078 × 10^−4^	4.926 × 10^−9^ **	−3.754 × 10^−4^	2.055 × 10^−10^ **	−3.962 × 10^−4^	1.122 × 10^−1^	−4.709 × 10^−4^
Lactose	2.561 × 10^−6^ **	7.373 × 10^−4^	5.423 × 10^−2^	2.917 × 10^−4^	2.201 × 10^−1^	8.632 × 10^−4^	2.578 × 10^−2^	4.988 × 10^−4^	6.792 × 10^−1^	9.006 × 10^−5^	6.042 × 10^−1^	5.276 × 10^−4^
Maltose	1.331 × 10^−14^ **	1.444 × 10^−3^	2.300 × 10^−12^ **	1.261 × 10^−3^	2.487 × 10^−2^	2.017 × 10^−3^	1.891 × 10^−11^ **	1.893 × 10^−3^	1.245 × 10^−8^ **	1.551 × 10^−3^	8.953 × 10^−3^	3.579 × 10^−3^
Non-milk extrinsic sugars	4.429 × 10^−93^ **	6.847 × 10^−4^	1.026 × 10^−58^ **	5.259 × 10^−4^	5.539 × 10^−11^ **	9.393 × 10^−4^	5.870 × 10^−48^ **	7.229 × 10^−4^	1.008 × 10^−31^ **	5.682 × 10^−4^	1.707 × 10^−6^ **	1.061 × 10^−3^
Other sugars	2.743 × 10^−15^ **	3.213 × 10^−3^	5.938 × 10^−9^ **	2.314 × 10^−3^	3.361 × 10^−2^	3.545 × 10^−3^	4.163 × 10^−15^ **	4.687 × 10^−3^	3.404 × 10^−11^ **	3.889 × 10^−3^	6.052 × 10^−2^	4.727 × 10^−3^
Sucrose	3.145 × 10^−132^ **	1.173 × 10^−3^	6.647 × 10^−86^ **	9.162 × 10^−4^	5.687 × 10^−15^ **	1.605 × 10^−3^	1.530 × 10^−57^ **	1.139 × 10^−3^	1.229 × 10^−37^ **	8.928 × 10^−4^	1.328 × 10^−7^ **	1.655 × 10^−3^

Ordinary least squares regression models were constructed for the two anhedonia items as outcomes, with each nutrient variable as a predictor in separate models, in addition to sex, age, body mass index, and Townsend index of socioeconomic deprivation as additional predictors in every model. Never-depressed and ever-depressed subgroups are defined according to the ICD-10 F32 or F33. **: significant at the Bonferroni-corrected *p* ≤ 2.55 × 10^−4^ significance threshold, PHQ9: Patient Health Questionnaire 9, ICD: International Classification of Diseases, F32 or F33: lifetime diagnosis of either a depressive episode or recurrent depressive disorder.

### 3.5. Relevance of Depression Phenotypes and Anhedonia Variables to Nutrient Intake as the Outcome

The *p*-value and unstandardized regression coefficient of each of the two lifetime depression diagnoses (Table 8, see also Table 1), the two current depression scores (Table 9, see also Table 1), and the two variables of current anhedonia level (Table 10, see also Table 1) as predictors in separate regression models are detailed for each nutrient variable as the outcome.

**Table 8 nutrients-17-02594-t008:** Results of the regression models of lifetime depression for nutrient intake as the outcome.

Outcome of the Regression Model	ICD-10 F32 or F33		CIDI Lifetime Depression	
	*p* Value	Coefficient	*p* Value	Coefficient
Energy	1.938 × 10^−9^ **	1.102 × 10^2^	1.665 × 10^−3^	5.985 × 10^1^
Energy density	1.803 × 10^−25^ **	1.270 × 10^−1^	3.309 × 10^−1^	1.244 × 10^−2^
Energy from beverages	5.565 × 10^−7^ **	3.385 × 10^1^	4.099 × 10^−1^	5.878
Total weight of beverages only	2.029 × 10^−24^ **	5.013 × 10^1^	1.907 × 10^−14^ **	3.988 × 10^1^
Carbohydrate	1.424 × 10^−21^ **	5.676	1.646 × 10^−7^ **	3.235
Free sugar	8.641 × 10^−51^ **	4.041	2.300 × 10^−9^ **	1.680
Fructose	2.011 × 10^−1^	1.422 × 10^−1^	1.283 × 10^−2^	2.930 × 10^−1^
Glucose	7.140 × 10^−4^	3.345 × 10^−1^	2.402 × 10^−4^ **	3.832 × 10^−1^
Intrinsic and milk sugars	4.179 × 10^−3^	5.918 × 10^−1^	4.391 × 10^−8^ **	1.214
Lactose	1.752 × 10^−29^ **	6.572 × 10^−1^	8.947 × 10^−13^ **	4.541 × 10^−1^
Maltose	2.529 × 10^−3^	1.476 × 10^−1^	6.309 × 10^−1^	2.427 × 10^−2^
Non-milk extrinsic sugars	2.487 × 10^−50^ **	4.091	1.084 × 10^−10^ **	1.846
Other sugars	1.608 × 10^−19^ **	2.039 × 10^−1^	6.038 × 10^−9^ **	1.383 × 10^−1^
Sucrose	4.934 × 10^−63^ **	3.206	6.077 × 10^−19^ **	1.772

Ordinary least squares regression models were constructed for each nutrient variable as the outcome, with each of the two lifetime depression diagnosis binary variables as a predictor in separate models, in addition to sex, age, body mass index, and Townsend index of socioeconomic deprivation as additional predictors in every model. **: significant at the Bonferroni-corrected *p* ≤ 2.55 × 10^−4^ significance threshold, ICD: International Classification of Diseases, F32 or F33: lifetime diagnosis of either a depressive episode or recurrent depressive disorder, CIDI: Composite International Diagnostic Interview.

**Table 9 nutrients-17-02594-t009:** Results of the regression models of current depression for nutrient intake as the outcome.

Outcome of the Regression Model	Four-Item Current Depression Score						PHQ9 Score					
	Total Sample		Never-Depressed Controls		Ever-Depressed Patients		Total Sample		Never-Depressed Controls		Ever-Depressed Patients	
	*p* Value	Coefficient	*p* Value	Coefficient	*p* Value	Coefficient	*p* Value	Coefficient	*p* Value	Coefficient	*p* Value	Coefficient
Energy	7.797 × 10^−64^ **	4.658 × 10^1^	1.438 × 10^−62^ **	5.326 × 10^1^	3.955 × 10^−2^	1.395 × 10^1^	2.466 × 10^−26^ **	1.960 × 10^1^	1.012 × 10^−22^ **	2.100 × 10^1^	5.304 × 10^−3^	1.201 × 10^1^
Energy density	1.266 × 10^−231^ **	5.949 × 10^−2^	3.873 × 10^−170^ **	5.877 × 10^−2^	1.186 × 10^−34^ **	5.544 × 10^−2^	1.283 × 10^−86^ **	2.445 × 10^−2^	2.636 × 10^−72^ **	2.581 × 10^−2^	2.170 × 10^−8^ **	1.642 × 10^−2^
Energy from beverages	5.085 × 10^−9^ **	5.949	8.860 × 10^−6^ **	5.253	6.274 × 10^−2^	4.456	1.007 × 10^−6^ **	3.382	2.609 × 10^−3^	2.421	2.452 × 10^−3^	4.782
Total weight of beverages only	6.804 × 10^−17^ **	6.179	5.266 × 10^−9^ **	5.010	1.486 × 10^−1^	2.542	5.304 × 10^−16^ **	4.094	2.219 × 10^−6^ **	2.777	1.769 × 10^−5^ **	4.986
Carbohydrate	2.255 × 10^−111^ **	2.007	4.186 × 10^−98^ **	2.172	1.089 × 10^−4^ **	8.512 × 10^−1^	2.054 × 10^−30^ **	6.870 × 10^−1^	1.307 × 10^−21^ **	6.637 × 10^−1^	1.262 × 10^−3^	4.516 × 10^−1^
Free sugar	6.335 × 10^−218^ **	1.277	4.588 × 10^−157^ **	1.246	5.439 × 10^−22^ **	9.791 × 10^−1^	4.585 × 10^−91^ **	5.518 × 10^−1^	6.367 × 10^−60^ **	5.146 × 10^−1^	2.015 × 10^−13^ **	4.798 × 10^−1^
Fructose	1.806 × 10^−10^ **	−1.067 × 10^−1^	3.372 × 10^−9^ **	−1.147 × 10^−1^	3.141 × 10^−4^	−1.444 × 10^−1^	1.006 × 10^−5^ **	−5.042 × 10^−2^	1.059 × 10^−7^ **	−7.056 × 10^−2^	8.371 × 10^−1^	−5.376 × 10^−3^
Glucose	5.928 × 10^−2^	−2.806 × 10^−2^	3.972 × 10^−2^	−3.543 × 10^−2^	2.524 × 10^−2^	−8.030 × 10^−2^	3.729 × 10^−1^	−9.021 × 10^−3^	3.603 × 10^−2^	−2.467 × 10^−2^	5.757 × 10^−1^	1.300 × 10^−2^
Intrinsic and milk sugars	3.670 × 10^−8^ **	−1.713 × 10^−1^	1.495 × 10^−7^ **	−1.892 × 10^−1^	1.922 × 10^−4^ **	−2.795 × 10^−1^	9.183 × 10^−4^	−7.129 × 10^−2^	4.363 × 10^−6^ **	−1.147 × 10^−1^	9.502 × 10^−1^	3.106 × 10^−3^
Lactose	4.248 × 10^−25^ **	9.081 × 10^−2^	1.629 × 10^−13^ **	7.497 × 10^−2^	3.647 × 10^−2^	4.404 × 10^−2^	7.955 × 10^−8^ **	3.312 × 10^−2^	1.046 × 10^−2^	1.831 × 10^−2^	3.478 × 10^−2^	3.060 × 10^−2^
Maltose	1.101 × 10^−13^ **	5.465 × 10^−2^	1.029 × 10^−12^ **	6.123 × 10^−2^	1.938 × 10^−1^	2.138 × 10^−2^	5.892 × 10^−14^ **	3.675 × 10^−2^	5.038 × 10^−11^ **	3.754 × 10^−2^	1.022 × 10^−2^	2.762 × 10^−2^
Non-milk extrinsic sugars	8.700 × 10^−202^ **	1.249	3.307 × 10^−144^ **	1.214	4.794 × 10^−20^ **	9.478 × 10^−1^	6.200 × 10^−86^ **	5.451 × 10^−1^	8.997 × 10^−56^ **	5.047 × 10^−1^	5.712 × 10^−13^ **	4.783 × 10^−1^
Other sugars	6.583 × 10^−42^ **	4.605 × 10^−2^	1.144 × 10^−29^ **	4.397 × 10^−2^	5.930 × 10^−4^	3.060 × 10^−2^	1.264 × 10^−39^ **	3.044 × 10^−2^	1.117 × 10^−28^ **	2.947 × 10^−2^	1.689 × 10^−5^ **	2.522 × 10^−2^
Sucrose	1.309 × 10^−278^ **	1.025	6.577 × 10^−200^ **	9.988 × 10^−1^	1.279 × 10^−28^ **	8.004 × 10^−1^	1.070 × 10^−111^ **	4.346 × 10^−1^	2.877 × 10^−72^ **	4.015 × 10^−1^	6.668 × 10^−17^ **	3.917 × 10^−1^

Ordinary least squares regression models were constructed for each nutrient variable as the outcome, with each of the two current depression scores as a predictor in separate models, in addition to sex, age, body mass index, and Townsend index of socioeconomic deprivation as additional predictors in every model. Never-depressed and ever-depressed subgroups are defined according to the ICD-10 F32 or F33. **: significant at the Bonferroni-corrected *p* ≤ 2.55 × 10^−4^ significance threshold, PHQ9: Patient Health Questionnaire 9, ICD: International Classification of Diseases, F32 or F33: lifetime diagnosis of either a depressive episode or recurrent depressive disorder.

**Table 10 nutrients-17-02594-t010:** Results of the regression models for anhedonia for nutrient intake as the outcome.

Outcome of the Regression Model	Anhedonia Item of the Four-Item Current Depression Questionnaire						Anhedonia Item of the PHQ9 Questionnaire					
	Total Sample		Never-Depressed Controls		Ever-Depressed Patients		Total Sample		Never-Depressed Controls		Ever-Depressed Patients	
	*p* Value	Coefficient	*p* Value	Coefficient	*p* Value	Coefficient	*p* Value	Coefficient	*p* Value	Coefficient	*p* Value	Coefficient
Energy	6.870 × 10^−9^ **	6.858 × 10^1^	5.367 × 10^−7^ **	6.710 × 10^1^	1.416 × 10^−1^	4.202 × 10^1^	7.059 × 10^−13^ **	7.006 × 10^1^	1.053 × 10^−10^ **	7.173 × 10^1^	6.500 × 10^−1^	1.084 × 10^1^
Energy density	1.217 × 10^−60^ **	1.306 × 10^−1^	6.105 × 10^−48^ **	1.305 × 10^−1^	2.226 × 10^−7^ **	1.008 × 10^−1^	2.014 × 10^−115^ **	1.480 × 10^−1^	1.753 × 10^−71^ **	1.317 × 10^−1^	1.827 × 10^−25^ **	1.660 × 10^−1^
Energy from beverages	4.479 × 10^−8^ **	2.427 × 10^1^	5.066 × 10^−5^ **	2.038 × 10^1^	6.095 × 10^−3^	2.872 × 10^1^	5.945 × 10^−13^ **	2.593 × 10^1^	3.812 × 10^−9^ **	2.426 × 10^1^	1.084 × 10^−2^	2.150 × 10^1^
Total weight of beverages only	1.419 × 10^−5^ **	1.406 × 10^1^	2.938 × 10^−2^	7.993	5.225 × 10^−2^	1.497 × 10^1^	2.635 × 10^−5^ **	1.101 × 10^1^	5.733 × 10^−2^	5.682	3.771 × 10^−1^	5.495
Carbohydrate	2.957 × 10^−11^ **	2.556	1.595 × 10^−7^ **	2.278	1.286 × 10^−1^	1.411	1.691 × 10^−30^ **	3.630	1.641 × 10^−23^ **	3.594	1.676 × 10^−1^	1.070
Free sugar	4.778 × 10^−52^ **	2.655	4.337 × 10^−35^ **	2.436	8.940 × 10^−7^ **	2.130	6.473 × 10^−104^ **	3.100	1.287 × 10^−66^ **	2.800	1.824 × 10^−12^ **	2.527
Fructose	2.123 × 10^−8^ **	−4.100 × 10^−1^	5.099 × 10^−9^ **	−4.848 × 10^−1^	1.477 × 10^−1^	−2.508 × 10^−1^	2.145 × 10^−12^ **	−4.161 × 10^−1^	5.360 × 10^−10^ **	−4.196 × 10^−1^	3.635 × 10^−5^ **	−5.846 × 10^−1^
Glucose	2.323 × 10^−3^	−1.977 × 10^−1^	2.406 × 10^−4^ **	−2.701 × 10^−1^	4.827 × 10^−1^	−1.080 × 10^−1^	3.781 × 10^−4^	−1.871 × 10^−1^	6.932 × 10^−4^	−2.036 × 10^−1^	3.757 × 10^−3^	−3.672 × 10^−1^
Intrinsic and milk sugars	4.926 × 10^−9^ **	−8.067 × 10^−1^	2.055 × 10^−10^ **	−9.926 × 10^−1^	1.122 × 10^−1^	−5.237 × 10^−1^	1.416 × 10^−9^ **	−6.672 × 10^−1^	5.762 × 10^−8^ **	−6.820 × 10^−1^	4.462 × 10^−5^ **	−1.079
Lactose	2.578 × 10^−2^	8.820 × 10^−2^	6.792 × 10^−1^	1.849 × 10^−2^	6.042 × 10^−1^	4.980 × 10^−2^	2.561 × 10^−6^ **	1.464 × 10^−1^	5.423 × 10^−2^	6.836 × 10^−2^	2.201 × 10^−1^	9.116 × 10^−2^
Maltose	1.891 × 10^−11^ **	2.107 × 10^−1^	1.245 × 10^−8^ **	2.034 × 10^−1^	8.953 × 10^−3^	1.866 × 10^−1^	1.331 × 10^−14^ **	2.005 × 10^−1^	2.300 × 10^−12^ **	2.099 × 10^−1^	2.487 × 10^−2^	1.306 × 10^−1^
Non-milk extrinsic sugars	5.870 × 10^−48^ **	2.590	1.008 × 10^−31^ **	2.351	1.707 × 10^−6^ **	2.107	4.429 × 10^−93^ **	2.982	1.026 × 10^−58^ **	2.669	5.539 × 10^−11^ **	2.392
Other sugars	4.163 × 10^−15^ **	1.163 × 10^−1^	3.404 × 10^−11^ **	1.099 × 10^−1^	6.052 × 10^−2^	7.286 × 10^−2^	2.743 × 10^−15^ **	9.482 × 10^−2^	5.938 × 10^−9^ **	7.873 × 10^−2^	3.361 × 10^−2^	6.664 × 10^−2^
Sucrose	1.530 × 10^−57^ **	1.985	1.229 × 10^−37^ **	1.791	1.328 × 10^−7^ **	1.642	3.145 × 10^−132^ **	2.487	6.647 × 10^−86^ **	2.264	5.687 × 10^−15^ **	1.986

Ordinary least squares regression models were constructed for each nutrient variable as the outcome, with each of the two anhedonia items as a predictor in separate models, in addition to sex, age, body mass index, and Townsend index of socioeconomic deprivation as additional predictors in every model. Never-depressed and ever-depressed subgroups are defined according to the ICD-10 F32 or F33. **: significant at the Bonferroni-corrected *p* ≤ 2.55 × 10^−4^ significance threshold, PHQ9: Patient Health Questionnaire 9, ICD: International Classification of Diseases, F32 or F33: lifetime diagnosis of either a depressive episode or recurrent depressive disorder.

## 4. Discussion

We investigated bidirectional associations of dietary energy, carbohydrate, and sugar subtype consumption with lifetime depression in a cross-sectional study in a European cohort—representing the largest study on sugar subtypes and depression so far. In addition, we calculated the associations of the same dietary parameters with current depression scores and anhedonia scores among depressed patients and also in never-depressed controls. Although controls had a higher intake of energy and energy from beverages, depressed patients still showed a higher energy density and total weight of beverages only. Interestingly, no difference was found in carbohydrate consumption between depressed persons and controls, but intake of almost all measured sugar subtypes showed significant differences between the two groups: daily intake of most sugar subtypes (free sugar, lactose, non-milk extrinsic sugars, other sugars, and sucrose) was elevated in depressed patients, while intake of three sugar types (fructose, glucose, and maltose) was higher in controls. Looking at the bidirectional associations, we observed almost identical results in both directions (i.e., the effect of diet factors on depression and vice versa). Associations between depression scores and energy, carbohydrate, and sugar intake were mostly positive. Interestingly, these positive associations were very similar in patients with a depression diagnosis and in never-depressed persons. Remarkably highly significant positive bidirectional associations in both patients and controls were revealed for energy density, free sugar, non-milk extrinsic sugars, and sucrose. Bidirectional associations between anhedonia scores and energy, carbohydrate, and sugar intake were in general also positive. Interestingly, these associations were also similar in patients and controls. Highly significant positive bidirectional associations of anhedonia scores in both patients and controls were found with energy density, free sugar, non-milk extrinsic sugars, and sucrose intake. The only general difference between these associations was that the regression coefficients in the direction from nutrient intake towards scores (either depression or anhedonia) were somewhat higher in depressed patients than in controls; while in the direction from scores towards intake, the coefficients were generally higher in controls than in patients. Furthermore, there were a few exceptions among the different sugar subtypes. Fructose showed consistently negative bidirectional associations with current depression and anhedonia, while intrinsic and milk sugars showed a positive bidirectional association with CIDI lifetime depression, but mostly negative associations with anhedonia and current depression. Bidirectional associations of maltose were also inconsistent: positive associations with current depression and anhedonia were detected only in the total sample and among controls, but not in the depressed subsample. Glucose also showed a positive bidirectional association with CIDI lifetime depression.

Our results are summarized in Table 1. In addition to the above summary, the table shows that the results for the different depression phenotypes are in high accordance: in most cases, the results for either or both of the lifetime depression variables show similar directions of effect as the two questionnaire-based current depression scores, bidirectionally (Table 1). So, all these results suggest that there are highly similar associations between nutrient intake and multiple depression variables, despite the differences in depression measures.

All our results were corrected for the effects of age, sex, BMI, and Townsend deprivation index.

The sex ratio was relatively well-balanced in our sample (females: 55.1%), which is a key novel feature of our study, as the previously published two studies [4,13] investigating the association between sugar subtypes and depression focused only on women. So, for the first time, associations between various sugar subtypes and depression/anhedonia can be generalized to both sexes. Nevertheless, future studies may address potential sex-related differences. On average, our participants are overweight according to BMI and non-deprived according to the Townsend deprivation index. According to ICD-10 criteria, 9.42%, while according to CIDI criteria, 14.75% showed depression—these results are comparable with the 12% lifetime prevalence of depression [1]. The NHS (https://www.nhs.uk/live-well/healthy-weight/managing-your-weight/understanding-calories/, accessed on 2 June 2025) recommends 2500 kcal a day for an average man and 2000 kcal a day for an average woman (although this should be adjusted according to age, weight, height, and exercise)—according to our results, neither controls (8644.42 kJ = 2066.06 kcal) nor depressed patients (8611.31 kJ = 2058.15 kcal) reached this average daily level. At the moment, recommendations for daily sugar intake are only restricted to free sugars, e.g., the NHS (https://www.nhs.uk/live-well/eat-well/food-types/how-does-sugar-in-our-diet-affect-our-health/, accessed on 8 May 2025) recommends a maximum intake of 30 g of free sugar a day. According to our results, the daily free sugar consumption of UK adults in the UKB is double the NHS recommendation and is also higher than the less strict WHO recommendation (50 g of free sugar for a 2000 kcal diet) [31], which corresponds to the mentioned overweight status of our sample. Depressed patients showed a significantly higher level of free sugar intake, but both depressed patients (63.19 g) and controls (60.65 g) consumed an unhealthy amount of free sugars. A previous UKB study by Kaiser et al. [32] detected that only free sugars in beverages (but not in solids) were associated with depression risk; furthermore, these associations depended on beverage subtypes. The authors highlighted that public health initiatives regarding depression risk should differentiate between free sugar and beverage subtypes. Based on our results, it should be considered to also add data on sugar subtypes to these recommendations, especially for restricting non-milk extrinsic sugars and sucrose.

The detected positive bidirectional associations between most of the measured dietary variables and depression generally support the known relationship between elevated energy and sugar intake and depression [10,12,14]. Furthermore, we also observed a similar bidirectional association with anhedonia. As we mentioned previously, the relationship between anhedonia and eating habits is more complicated; however, our positive results are in line with studies showing a higher consumption of sweetened beverages among people with anhedonia [8] and the predictive role of anhedonia for weight-gain in the long term [9]. Previous studies highlighted various mechanisms to explain the depressogenic effects of elevated sugar consumption, including (1) the increased level of circulating inflammatory markers through insulin secretion, and (2) the subsequent decrease of BDNF; through (3) somatic conditions comorbid with depression, including obesity, type II diabetes, and cardiovascular disease; (4) alterations in the gut microbiome and gut–brain circuits; and (5) increased oxidative stress [13,33,34,35]. Furthermore, as our results are cross-sectional, increased carbohydrate intake may be a consequence of craving related to self-regulation for depressive mood. If we compare our bidirectional regression results, we can see that the directions of the effects and the significance values are highly similar; at the same time, the effect sizes are generally much higher in the case of testing the effect of depression and anhedonia on dietary factors compared to the other direction—suggesting that depression and anhedonia might have a higher impact on sugar intake than vice versa, even though the interpretability of unstandardized regression coefficients is limited when comparing effect sizes between different variables. However, it has to be noted that the intake of most sugar subtypes was higher among depressed patients, but the differences were usually below 1 g (with a maximum of 2.5 g in the case of free sugar) between the controls and depressed patients in our sample. At the same time, positive associations between dietary factors and depression/anhedonia symptoms were also replicated in the control group, representing a general population from a high-income country. Therefore, our results might suggest that easy access to unlimited sweetened beverages and foods may serve as a common “solution” for many, especially in case of experiencing low mood or anhedonia (even at subclinical levels). The recurrent stimulation of the reward system by high-energy, sugary foods and beverages can promote addictive-like behaviors [36] and lead to diseases of civilization, such as type II diabetes and obesity. The detected high consumption of free sugars and overweight status of our sample may represent a similar phenomenon.

Regarding sugar subtypes, sucrose, free sugar, non-milk extrinsic sugars, other sugars, lactose, and maltose showed generally positive bidirectional associations with depression and anhedonia. Glucose showed a positive bidirectional association only with CIDI, but not ICD-10, lifetime depression. A previous study similarly detected a positive association for free sugars, sucrose, and glucose with self-reported depression symptoms among young Japanese women, although lactose showed no effect [13]; while a US study also revealed a risk effect of added sugar (which is similar to free sugar), but a protective effect of lactose, and only an elevated but non-significant effect of sucrose and glucose, on self-reported depression symptoms among post-menopausal women [4]. As we already mentioned, the UKB study by Kaiser et al. also supported the risk effect of free sugars from beverages on ICD-10 (F32/33) depression [32]. However, the same study found no relationship between intrinsic sugars and depression risk. We similarly did not find an association between intrinsic and milk sugars and ICD-10 lifetime depression, but interestingly detected a positive association with CIDI lifetime depression and a negative association with current depression and anhedonia, bidirectionally. At the moment, we are not able to explain these conflicting results. A higher consumption of intrinsic and milk sugars might be a health-conscious choice even by people with lifetime depression (nevertheless, our study did not show differences in the intake of intrinsic and milk sugars between depressed patients and controls). Intrinsic sugars are found in fruits, vegetables, and milk [31], and a meta-analysis detected a significant protective association of fruit and vegetable intake with depression [37].

We also found bidirectional negative associations between fructose and current depression and anhedonia symptoms, while the mentioned Japanese study [13] detected a positive association with self-reported depression among young females, and the US study [4] found no effect. The contradictory results might be explained by the fact that fructose is naturally found in fruits and honey, but is also frequently added to various foods and beverages, especially those targeted for those with type II diabetes, but with probably less beneficial effects on overall health.

Our results suggest that more studies are needed to explore the relationship between dietary intake of sugar subtypes and depression, as well as specific depression symptoms, such as anhedonia. We found no differences in carbohydrate intake between depressed patients and controls but detected mostly higher consumption of sugar subtypes in the depressed group; furthermore, while most sugars showed a positive association with depression and anhedonia, in some cases negative associations were also captured. These results suggest that studies should include detailed data on the intake of different sugar subtypes. Since diet is a modifiable factor, advanced recommendations for daily energy intake and intake of different sugar subtypes could be beneficial for depression treatment and prevention. Additionally, our results among the controls also suggest that recommendations and education on health-conscious dietary choices are also important in the general population.

### Limitations

The Oxford WebQ dietary questionnaire, being a 24 h recall questionnaire, may suffer from recall bias, and thus may be prone to some systematic biases, such as underreporting by more depressed individuals. However, former validation analyses with objective biomarkers as a standard have revealed that this self-report questionnaire performs well in comparison with interviewer-based 24 h recall measurement [25].

As a cross-sectional study, we were not able to capture causal associations. In the case of diet and depression, both effect directions are possible (i.e., the depressogenic factors of dietary factors and disadvantageous changes in diet as a result of depression). In the case of the subgroup analyses, almost all results showed the same direction, but many of them did not survive Bonferroni correction in the depressed group—this could partly be explained by the much lower sample size of this group compared to controls. In general, Bonferroni correction may increase the possibility of false negative results in our present study. However, this most stringent form of correction can control the number of false positives within the high number of tests.

Another limitation of our analyses could be the imbalanced rate between cases and controls in both lifetime depression binary phenotypes (Table 2), entailing some potential bias in the results of the logit regression models with either of these two binary variables as the outcome (Table 5). Nevertheless, Table 1 suggests the same pattern in significance and direction of effect for all these nutrient–depression variable pairs between the two directions, and thus the same pattern in ordinary least squares regression models with lifetime depression as a predictor and nutrient as an outcome.

Anhedonia was captured as an item on depression questionnaires, and future studies with separate anhedonia measurements are needed to strengthen our results. However, we used a large sample, and for the first time showed bidirectional associations of various sugar subtypes with depression and anhedonia in a well-balanced sample of both sexes. Furthermore, we also included multiple depression phenotypes to our analyses to capture dietary associations with both diagnosed and subclinical levels of depression.

Since the UK Biobank database may not be representative of the entire UK population, future replication studies will be needed to strengthen our present results within the UK. Moreover, only replication studies within a diverse range of distinct populations and countries will be able to determine the robustness of our present results in the future.

## 5. Conclusions

In conclusion, our results support the known association of elevated energy and sugar intake with depression, and, as novel evidence, with anhedonia, one of the core symptoms of depression. Among more general dietary factors, the total weight of beverages only, energy density, and carbohydrate, whilst among sugar subtypes, free sugars, non-milk extrinsic sugars, and sucrose showed the strongest bidirectional associations with depression. In the case of fructose, we detected a protective association with depression and anhedonia symptoms. Based on our results and future studies, it should be considered to complement dietary recommendations for depression with data on sugar subtypes.

## Figures and Tables

**Table 1 nutrients-17-02594-t001:** Summary of all results, including the bidirectional association between depression/anhedonia and nutrient intake.

	Significant Difference Between Depressed Patients and Controls	Depression								Anhedonia Item					
		ICD-10 F32 or F33 Depression	CIDI Lifetime Depression	Four-Item Current Depression Score			PHQ9 Score			Of the Four-Item Current Depression Questionnaire			Of the PHQ9 Questionnaire		
				Total Sample	Never- Depressed Controls	Ever- Depressed Patients	Total Sample	Never- Depressed Controls	Ever- Depressed Patients	Total Sample	Never- Depressed Controls	Ever- Depressed Patients	Total Sample	Never- Depressed Controls	Ever- Depressed Patients
Energy	↓	↑ ↑	n n	↑ ↑	↑ ↑	n n	↑ ↑	↑ ↑	n n	↑ ↑	↑ ↑	n n	↑ ↑	↑ ↑	n n
Energy density	↑	↑ ↑	n n	↑ ↑	↑ ↑	↑ ↑	↑ ↑	↑ ↑	↑ ↑	↑ ↑	↑ ↑	↑ ↑	↑ ↑	↑ ↑	↑ ↑
Energy from beverages	↓	↑ ↑	n n	↑ ↑	↑ ↑	n n	↑ ↑	n n	n n	↑ ↑	↑ ↑	n n	↑ ↑	↑ ↑	n n
Total weight of beverages only	↑	↑ ↑	↑ ↑	↑ ↑	↑ ↑	n n	↑ ↑	↑ ↑	↑ ↑	↑ ↑	n n	n n	↑ ↑	n n	n n
Carbohydrate	n	↑ ↑	↑ ↑	↑ ↑	↑ ↑	↑ ↑	↑ ↑	↑ ↑	n n	↑ ↑	↑ ↑	n n	↑ ↑	↑ ↑	n n
Free sugar	↑	↑ ↑	↑ ↑	↑ ↑	↑ ↑	↑ ↑	↑ ↑	↑ ↑	↑ ↑	↑ ↑	↑ ↑	↑ ↑	↑ ↑	↑ ↑	↑ ↑
Fructose	↓	n n	n n	↓ ↓	↓ ↓	n n	↓ ↓	↓ ↓	n n	↓ ↓	↓ ↓	↓ n	↓ ↓	↓ ↓	n ↓
Glucose	↓	n n	↑ ↑	n n	n n	n n	n n	n n	n n	n n	n ↓	n n	n n	↓ n	n n
Intrinsic and milk sugars	n	n n	↑ ↑	↓ ↓	↓ ↓	↓ ↓	n n	↓ ↓	n n	↓ ↓	↓ ↓	↓ n	↓ ↓	↓ ↓	n ↓
Lactose	↑	↑ ↑	↑ ↑	↑ ↑	↑ ↑	n n	↑ ↑	n n	n n	↑ n	n n	n n	n ↑	n n	n n
Maltose	↓	n n	n n	↑ ↑	↑ ↑	n n	↑ ↑	↑ ↑	n n	↑ ↑	↑ ↑	n n	↑ ↑	↑ ↑	n n
Non-milk extrinsic sugars	↑	↑ ↑	↑ ↑	↑ ↑	↑ ↑	↑ ↑	↑ ↑	↑ ↑	↑ ↑	↑ ↑	↑ ↑	↑ ↑	↑ ↑	↑ ↑	↑ ↑
Other sugars	↑	↑ ↑	↑ ↑	↑ ↑	↑ ↑	n n	↑ ↑	↑ ↑	↑ ↑	↑ ↑	↑ ↑	n n	↑ ↑	↑ ↑	n n
Sucrose	↑	↑ ↑	↑ ↑	↑ ↑	↑ ↑	↑ ↑	↑ ↑	↑ ↑	↑ ↑	↑ ↑	↑ ↑	↑ ↑	↑ ↑	↑ ↑	↑ ↑

Table 1 summarizes the effects and directions of all analyses. In the case of the bidirectional regression models, each cell represents two models, and the left arrow represents the result of the model, with nutrient intake as a predictor and depression/anhedonia as an outcome; while the right arrow represents the result of the model with depression/anhedonia as a predictor and nutrient intake as an outcome. An arrow means that the given result is significant after Bonferroni correction, n: not significant after Bonferroni correction. If both models survived Bonferroni correction, the color shade is darker; if only one of the two models survived Bonferroni correction, the color shade is lighter. ↑/orange: significantly higher in depressed patients, or significant positive association after Bonferroni correction; ↓/green: significantly lower in depressed patients, or significant negative association after Bonferroni correction.

## Data Availability

The data presented in this study are available on request from the corresponding author. The data are not publicly available, due to ethical considerations.

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
