# Peer review of "Consumption of High-Energy Food and Sugar Shows a Strong Positive Association with Low Mood in Control Subjects and Depressed Patients"

_nutrients, 2025, doi:10.3390/nu17162594_

Round 1
Reviewer 1 Report
Comments and Suggestions for Authors
Consumption of high-energy food and sugar shows strong positive association with low mood in control subjects and depressed patients
This study provides a valuable investigation into the bidirectional relationship between high-energy foods and specific sugar subtypes and depression/anhedonia symptoms. Using the robust UK Biobank dataset, the authors compare dietary patterns between individuals with and without depression diagnoses, offering important insights into how different types of sugars may relate to mental health outcomes. However, several aspects of the study's presentation and methodology could benefit from clarification.
Comments:
- The introduction establishes the significance of studying diet-depression relationships well, but could more sharply articulate what makes this particular investigation novel. While previous research has examined broad categories like "added sugars," this study's focus on specific sugar subtypes represents an important advance - but this distinction could be highlighted more explicitly. Similarly, the rationale for examining bidirectional effects could be strengthened by more directly addressing whether the field currently debates the directionality of these relationships.
- The phrase “anhedonia item of the four-item depression questionnaire” vs. “anhedonia item of PHQ9 depression questionnaire” may confuse readers. Are these two anhedonia items conceptually the same but from different instruments?
- Please clarify how you handled discordance between diagnostic classification (ICD-10 ever/never-depressed) and current symptom scores (PHQ-9 or four-item depression score). For example, were individuals with no ICD-10 depression diagnosis but high PHQ-9 scores retained in the “never-depressed” group? Explicitly stating this in the Methods section would improve clarity and reproducibility. If someone is classified as “never-depressed” using the ICD-10 diagnosis, but shows depressive symptoms on the PHQ-9 or the four-item depression score, this raises an important methodological issue.
- The presentation of results, particularly in Table 1, includes some confusing elements. The meaning of the grey shading isn't immediately apparent and would benefit from explanation in the legend or caption. Additionally, the extremely precise p-value threshold (p ≤ 0.0035714286) seems unnecessarily specific - rounding to three significant figures or using standard scientific notation would improve readability without sacrificing precision.
Reviewer 2 Report
Comments and Suggestions for Authors
- The dietary data stem from the Oxford WebQ, a self-administered 24-hour recall measure, which is prone to recall bias and does not reflect usual intake. Averaging across multiple time points improves reliability but cannot eliminate systematic bias (e.g., underreporting by depressed individuals).
- The use of multiple, partially overlapping definitions of depression (ICD-10, CIDI, PHQ-9, 4-item score) is methodologically reasonable, but the diagnostic accuracy and clinical comparability across these tools are not discussed.
- The UK Biobank is known for a healthy volunteer selection bias. The analytic sample excludes those with missing dietary or depression data, which may further skew representativeness. Some literature has raised concerns that the UK Biobank is not representative of the diversity of the UK population or is not applicable to diverse populations (limitation).
- Table 1. For clarity, the authors should indicate in the title that this table was extracted from the UK Biobank, as indicated in section 2.2.
- The authors should consider not reporting more decimal places than the precision of the data or measurement allows. Typically, most manuscripts report 2 or possibly 3 decimal places.
- Neglect of mediation/moderation modeling: Although the authors discuss bidirectional relationships and potential psychological mechanisms, no mediation (e.g., does anhedonia mediate sugar-depression links?) or moderation (e.g., sex, BMI, SES interactions) analyses are conducted.
- The authors used logistic regression on a large sample (n > 100,000), which reduces the risk of convergence issues, but they did not acknowledge or address potential bias due to outcome imbalance. Given that ~90% of the sample was non-depressed, the CIDI and ICD-10 models may yield inflated odds ratios or misleading significance, particularly for sugar subtypes with small effect sizes. The authors may wish to explore Firth’s penalized likelihood logistic regression or Bayesian analysis.
- Use of unstandardized coefficients. Tables present raw coefficients (e.g., “0.00003” for energy intake), which are not interpretable without context. Standardized coefficients or effect sizes (e.g., odds ratios) would better communicate practical significance. Also reporting 95% confidence intervals would be more informative than these adjusted p-values..
- Table 3. The authors should provide unit abbreviations in the footnotes for ease in reading the table.
- Table 3. Looking at the binary depression measures, one might be concerned about the possibility of an overinflated distribution for the 4-item and PHQ9 measures, possibly questioning the use of OLS regression.
- Table 5, the authors should indicate these are logistic regression models, and the parameters of logits. This table is confusing, your text indicates that these are separate regression models per predictor, but the table typically shows only two models with multiple predictors, confusing.
- Table 8. Confused about this table, are these parameters from 28 separate models? The way the table is setup, similar to table 5, will confuse readers.
- The authors seemed to have run hundreds of models and applied Bonferroni correction. While this controls false positives, the sheer volume of comparisons may inflate false negatives and raise concerns about data dredging.
- I am assuming the authors used Python to write their regression routines, which they should specify. Also, it seems that they may have used ChatGPT-4 in constructing their code? While the authors did not use the typical statistical packages (e.g., SAS, Stata, etc.) one always is concerned about algorithmic accuracy. This has been a concern in pharmacological research with the use of R.
Round 2
Reviewer 1 Report
Comments and Suggestions for Authors
The authors have provided clear and thorough responses to my comments. I recommend the manuscript for publication.
Author Response
Thank you a lot for your time and also for your approval.
Reviewer 2 Report
Comments and Suggestions for Authors
- Table 1. I would consider this a supplemental table, it is not your typical table 1 describing your study sample. Looks like table 3 and 4 should be table 1. The current table 1 really doesn’t provide the reader with crucial information about the sample, only referencing where data were obtained and during which time periods.
- My previous comment; “The authors should consider not reporting more decimal places than the precision of the data or measurement allows. Typically, most manuscripts report 2 or possibly 3 decimal places.” This should apply to tabled values, especially coefficients.
- Tables must independently convey information and enhance readers' understanding of results. While Table 2 efficiently conveys complex information, augmented by the parameter tables, it may pose a difficulty to readers.
- Table 2. The footnote indicates “. . . the first sign represents the results. . .” what is the sign, do you mean arrow? If arrows mean sign, then I understand the direction and multiple arrows.
- Table 2. What does “ n n “ stand for? Statistically adjusted non-significance?
- Table 2. Again, each cell represents two regression models, with the first column indicating predictor variables and the columns indicating outcomes.
